# Sustainability of the Adjustment Schemes in China's Grain Price Support Policy—An Empirical Analysis Based on the Partial Equilibrium Model of Wheat

**Jingdong Li [1,2], Weidong Liu [1,2]**  **and Zhouying Song [1,2,*]**

[1] Institute of Geographic Sciences and Natural Resources Research, Chinese Academy of Sciences, Beijing 100101, China; lijingdong@igsnrr.ac.cn (J.L.); liuwd@igsnrr.ac.cn (W.L.)

[2] Key Laboratory of Regional Sustainable Development Modeling, Chinese Academy of Sciences, Beijing 100101, China

* Correspondence: songzy@igsnrr.ac.cn

**Abstract:** The minimum purchase price policy for wheat and rice implemented by the Chinese government has achieved the fundamental goals of stabilizing grain prices, promoting production, and ensuring food security. This policy has also had negative impacts such as domestic and foreign price spreads and continuous increases in stocks and imports, which are not conducive to China's grain security development and thus unsustainable. Therefore, this paper builds a partial equilibrium model of China's grain market by simulating the effects of canceling or reducing the minimum purchase price on the market price, production, consumption, stock, and net import of wheat and then evaluates the sustainability of various adjustment programs. The research results show that first, lowering the minimum purchase price of wheat can reduce the domestic and foreign price spread, stock, and imports to a certain extent; however, it cannot fundamentally solve the negative impact of this policy. Second, cancellation of the minimum wheat purchase price policy can significantly reduce domestic and foreign price spread, stock, and imports; however, it will also significantly reduce wheat production and threaten China's grain security. Third, cancellation of the minimum wheat purchase price and the increase in agricultural production subsidies can solve the negative impact of the minimum purchase price policy and reduce the impact of the cancellation of the minimum purchase price policy on grain supply security. This policy adjustment is more sustainable than China's current policy. Finally, this paper asserts that China's grain price policy reform will influence and have implications for stakeholders in the global grain industry.

**Keywords:** wheat price; policy adjustment; sustainability; partial equilibrium model

## 1. Introduction

China is the world's largest grain producer and consumer, and because of its population size, ensuring grain self-sufficiency is crucial. To ensure food supply security, the Chinese government set a policy intervention objective that the grain supply should be dominated by domestic production and supplemented by imports [1]. The minimum support price policy was implemented to promote the policy objectives [2] by raising farmers' income expectations, which would encourage them to grow food. Under the national protecting policy, China's grain output was 615.21 million tons in 2017, an increase of 209.98 million tons or 51.82% over 2000. The strategic goal of ensuring food supply security has thus been achieved. However, changes in domestic and international environments pose a challenge to China's agricultural policy intervention system [3]. In the international grain market, the financial and energy attributes of grain are increasingly prominent (With the continuous development of

global financial markets, the impact of monetary liquidity, futures prices and international speculation on grain prices has gradually increased. The phenomenon that grain price fluctuations are linked to financial market and economic development is called grain financialization, also known as the financial attributes of grain [4]. The rapid development of international bioenergy has increased grain demand, impacted the balance of global grain supply and demand, and made the grain market associated with the development of biomass energy. This attribute is called the energy attribute of grain [5]). Multinational companies promote the global flow of grain by allocating and hedging grain worldwide [6]. However, because of the minimum purchase price policy of China's grain, the domestic grain price has risen rigidly and higher than that price abroad [7], and redundant stock has increased; thus, the implementation of the policy has become less sustainable. According to the statistics from the Department of Market and Economic Information of the Ministry of Agriculture, the prices of domestic wheat, corn, rice, and other major grain have been higher than the import price. The price difference between domestic and foreign countries has widened sharply and has remained high since the second half of 2013. In the context of frequent global trade friction, domestic grain prices are higher than foreign grain prices, and this puts China in a disadvantageous position. Additionally, the continuous increase of net food imports has triggered international public opinion and aroused the Chinese government's awareness of policy reform.

With the increase in per capita income in China, China's demand for grains such as wheat and rice have been partly replaced by the demand for meat, poultry, dairy products, vegetables, fruit, and aquatic products [8,9]. The focus of the Chinese government's policy has gradually shifted to optimizing the mix of grain varieties and ensuring quality and safety. In this context, the Chinese government's system of agricultural market intervention policies to ensure the supply and quantity of agricultural products has not adapted to its governance concept of sustainable development. Therefore, policy transformation is a new problem that the Chinese government must resolve [10].

In order to ensure the security of food supply and the benefits of grain farmers, China has issued a series of policies to ensure and support grain production, the most important of which are grain price support policies and grain production subsidies. In terms of food price support policies, the Chinese government has implemented the minimum purchase price policy for rice and wheat, and the temporary state collection and storage policy for corn (abolished in 2016). In 2004, the Chinese government formulated the current implementation plan for the minimum purchase price of rice based on the planting costs and reasonable profits (15–20% of the price) of the previous year and announced the minimum purchase price before rice planting to ensure farmers' enthusiasm. After the rice was listed in 2004, the purchase price in the regions where the plan had been implemented was lower than the previously established minimum purchase price for rice, China grain reserves group LTD company purchased rice according to the minimum purchase price. In 2006, on the basis of the successful implementation of the minimum purchase price policy for rice, the Chinese government began to implement the minimum purchase price policy for wheat. In 2008, in order to guarantee the corn production, the Chinese government implemented a temporary purchase and storage policy of corn. From the perspective of grain production subsidy policy, since 2003, China has successively implemented direct grain subsidies, comprehensive subsidies for agricultural materials, improved seed subsidies, and agricultural machinery purchase subsidies. After 2016, direct subsidies for grain growing, comprehensive subsidies for agricultural materials, and subsidies for improved varieties have been merged into agricultural support and protection subsidies. After the cancellation of the temporary purchase and storage policy for corn, the minimum purchase price policy for rice and wheat and grain production subsidies are important policies affecting China's grain production. The minimum purchase price policy has caused a distorting effect on grain production, purchasing and storage, and trade by increasing the market price of grain, and has destroyed the original market mechanism. Grain production subsidies mainly affect grain production. At this stage, China's grain production subsidies are not strong, so the impact of grain production subsidies on the grain market is much smaller than the minimum purchase price policy.

Since 2004, successive implementation of the minimum support price policy of wheat, rice, and corn has realized the basic goal of stabilizing grain prices and increasing production and farmers' income. However, the continuous increase in the minimum purchase price and the temporary state collection and storage price has also had a greater negative impact: Break grain price balance between China and abroad and increase the grain storage burden and national financial burden. In recent years, China's grain planting costs have increased rapidly. In order to ensure farmers' income from growing grain and the security of grain supply, the minimum purchase price implemented is higher than the market equilibrium price. The excessive prices lead to oversupply in the market, and the remaining grain production is purchased by the state. In order to increase the market price of grain, the minimum purchase price and the temporary state collection and storage price continue to increase, and the state's policy-related purchases have also continued to increase, which has increased the pressure on grain stocks and the national financial burden. At the same time, the rise in grain market prices has widened the gap between domestic and foreign grain prices and increased the pressure on grain imports. After 2013, the domestic prices of wheat, rice, and corn gradually exceeded their dutiable import prices within quota, and close to their dutiable import prices out of quota, leading to a decline in grain exports and an increase in imports. To reduce the negative impact of the minimum support price policy, guarantee national food security, and improve the sustainability of policy implementation, the Chinese government proposed "agricultural supply-side structural reform." The reform is supposed to solve the problem of the coexistence of the structural surplus and structural shortage of China's grain supply, which is a positive signal in policy reform. The 2016 Government Work Report of China emphasized establishing a market-based mechanism for the formation of grain prices and that marketization is the direction of the reform of grain policies. In recent years, China has made adjustments to its grain price policy. The purpose is to reduce the negative impacts such as domestic and foreign price spreads, continuous increases in stocks and imports, and to explore adjustment options with higher sustainability. For example, the minimum support price policy of corn was lowered in 2015 and replaced by a target price policy in 2016, and the minimum purchase price of wheat was unchanged from 2014 to 2017 and was appropriately lowered in 2018. The minimum purchase price of early indica rice has been lowered since 2016, and the minimum purchase price of medium and late indica rice and japonica rice has been lowered since 2017. Promoting market-oriented reform of the purchasing and storage policy was the direction of the minimum purchase price of wheat, and the yield of wheat is critical to national security. Furthermore, there is a debate on the implementation of the reform, which slows the pace of the adjustment of grain policy.

Additionally, wheat is one of the most important grain varieties in China, and the minimum purchase price policy of wheat is also the main component of China's grain price support policy. Hence, it is of great practical significance to study the policy reform of the minimum purchase price of wheat to explore the direction of China's grain price support policy reform. In this study, we build the partial equilibrium model of China's grain market, which can be used to analyze China's grain market operation mechanism and the impact of different reformation plans, including gradually reducing the minimum purchase price, only canceling the minimum purchase price policy, and a combination of canceling the minimum purchase price policy and greatly increasing agricultural subsidies. The research results are helpful in solving the dispute over the reformation of China's grain price policy. On the premise of ensuring the security of China's grain supply, choosing a reform plan that is in line with China's national conditions and is highly sustainable is necessary. Furthermore, this study provides international grain enterprises with new insights into China's policy trends and making reasonable decisions and other developing countries with experiences to apply to grain policy reforms.

## 2. Literature Review

The existing research about the grain price support policy mainly focuses on the evaluation of the policy effect and their reformation direction. On one hand, the minimum support price policy has a positive impact, it has stabilized the grain price [11,12] and promoted the continuous expansion

of grain planting area [13,14], continuous increased of grain output and significant improved grain productivity [15–18]. Price support policy can also effectively improve the income and welfare level of farmers and producers [19,20]. On the other hand, minimum support price policy has a negative impact. Firstly, the minimum support price policy distorts the market information that leads to the misallocation of resources, which causes the surplus of grain production and the imbalance of market supply and demand [21–24]. Secondly, when the domestic grain price rises substantially, it is forced to raise the minimum support price, which brings a great burden to the national finance [25,26]. Thirdly, the rise of grain prices reduces the level of consumer's welfare in the short term, while the continuous increase of grain production in the long term narrows the room for rising of the minimum support price and damaging farmers' welfare [27]. In recent years, the United States, Europe, South Korea, and other major developed countries and regions have actively promoted the reform and transformation of their purchase policies in order to minimize the use of policies and measures that seriously distort the market. In terms of improving the minimum support price policy, the existing research mainly gives two sets of plans. The first plan is to abolish the minimum purchase price policy and implement target price subsidy [10,28,29]. The second plan is to adopt a combination of reforms such as lowering the level of support prices and subsidies on grain yields or insurance on target prices [30].

Regarding grain production and food security research, scholars have focused on, for example, grain price fluctuation, policy formulation, resource utilization, climate change, economic development, and biodiversity protection [31–35]. Relatively few analyses have assessed the sustainability of food policy implementation. In recent years, research on the sustainability of policy implementation has gradually attracted the attention of scholars. From the perspective of the sustainability of policy implementation, food policy affects food production, and establishing a complete sustainable food system is a critical means to ensure food security [36]. A sustainable food policy means changing policy objectives, policy frames, policy mixes, and evaluation approaches. Only by exploring sustainable food policies can society effectively manage food and nutrition security, natural resource protection, and social justice [37]. Many empirical studies have been conducted on the sustainability of policy implementation, for example, (1) cointegration test analysis through the cointegration test of the time series of policy-related variables, to determine the sustainability of policies implemented by the government [38,39]; (2) Granger causality test analysis by building a panel Granger causality model to analyze sustainable policy systems, which has played an important role in achieving ecological protection and economic development [40]; and (3) compound index analysis to establish a forward-looking index system by using the data and information to evaluate the sustainability of future policies [41,42].

The aforementioned research has certain reference significance to conduct this study on grain price support policy reformation. However, the aforementioned research neither discusses the impact of different policy reformation plans on the development of Chinese grain based on China's national conditions nor conducts a sustainability analysis of various policy adjustment plans, which leads to disagreements between Chinese scholars and policymakers on the choice of reform plans for the grain price support policy. This paper holds that it is essential to clarify the mechanism of the impact of different policy reformation plans on the domestic grain market and national grain security to solve the dispute. Additionally, the empirical analysis methods have limitations: The cointegration test and Granger causality test cannot fully explore the impact of policy implementation on various parts of the economic system. This paper builds a partial equilibrium model of China's grain market. Through the estimation values of equation parameters, we simulate the impact of various adjustment plans of grain price support policies on the market price of wheat, production, consumption, stock, and net import. Combining with the goals of market-oriented reform of China's grain control policies, the sustainability of the implementation of various adjustment plans is evaluated in terms of reducing domestic and foreign price differences, stocks, and imports. This paper enriches the sustainability research on China's grain price support policy adjustments and provides bases for the selection of purchase price policy reform plans.

## 3. Data and Methods

To explore the impact of policy adjustment methods on China's grain market, we first analyze the operating mechanism of the Chinese grain market. The local equilibrium model based on the theories of Labys [43] and Smith [44] has been widely used in market operation mechanism research. Therefore, before analyzing the impact of China's grain price support policy adjustment, this paper first constructs a partial equilibrium model of China's grain market and quantitatively analyzes the operation mechanism of China's grain market.

### 3.1. Partial Equilibrium Model of the Grain Market

Under the intervention of China's current grain policy, the grain supply and demand system is jointly determined by the market mechanism and national grain policy and affected by changes in the economic environment and external shocks. The market mechanism has two aspects: The total supply of grain and the total demand for grain. The total supply comprises domestic grain output and grain imports. The total demand comprises domestic grain consumption and grain exports [43]. Referring to the analysis method of Lord [45], the total supply of grain and the total demand can be divided into four parts: Production, consumption, trade, and inventory. Changes in the economic environment and external shocks also affect grain supply and demand, especially the gradual enhancement of the financial attributes of grain, which promotes the global flow of resources and has a greater impact on grain supply and demand in the world [4,6]. Considering China's grain import quota policy, the impact of grain financialization on China's grain imports and exports is small, and the impact of financialization on grain is more formed from financial channels than from the process of grain production and trading [46]. The gradual enhancement of the financial attribute of grain aggravates the grain price fluctuation to a certain extent, impacts the balance of grain supply and demand, and thus jeopardizes China's food security [47]. Therefore, when constructing the partial equilibrium model of grain, the analysis is based on the market supply and demand theory, and the influence of national grain policies and external transmission factors on grain prices is considered. Therefore, this paper divides the partial equilibrium model of grain into a production module, consumption module, trade module, inventory module, and price transfer module for analysis (Figure 1).

### 3.1.1. Production Module

In the grain planting area equation, the grain growers make an expectation based on the previous grain purchase price and then adjust the grain planting area in the current period while also being affected by the previous planting area [48]. The national grain subsidy policy and price support policy increase the grain planting area by improving the enthusiasm of the grain growers [49]. The Chinese government formulates the current implementation plan for the minimum purchase price based on the planting costs and reasonable profits of the previous year. After new grains are listed, if the farmer's selling price in the planned implementation area (major grain-producing province) is lower than the minimum purchase price announced that year, the government uses the minimum purchase price to purchase new grains to achieve the goal of increasing the grain market price. If the grain purchase price in the planned implementation area is lower than the minimum purchase price announced in the year, the minimum purchase price policy will not be activated. China's grain production subsidies ensured farmers' enthusiasm for planting and increased grain planting area to a certain extent. The price of each input factor of grain planting is also an important factor that affects the area of grain planting. When the price of an input factor rises significantly, it increases the cost of grain production, and the profit of grain planting decreases sharply, which suppresses the enthusiasm of the grain growers and is not conducive to the increase in grain planting area. Therefore, the grain planting area can be expressed as:

$$\log A_{it} = \alpha_0^A + \alpha_1^A \log P_{i,t-1}^u + \alpha_2^A \log P_{it}^{gov} + \alpha_3^A \log A_{i,t-1} + \alpha_4^A \log SUP_{it} + \sum_j \beta_j^A (\log P_{jt}^s) + \varepsilon_{it}^A \quad (1)$$

where $A_{it}$ represents the planting area of grain $i$; $P_{it}^u$ represents the farmer's selling price of grain $i$; $P_{i,t-1}^u$ is lagged farmer's selling price of grain $i$; $P_{it}^{gov}$ represents the minimum purchase price; $SUP_{it}$ represents the amount of grain subsidies; $P_{jt}^s$ represents the price of input element $j$; $\alpha_0^A$ and $\varepsilon_{it}^A$ represent constant term and random error term, respectively; $\alpha_1^A, \alpha_2^A, \alpha_3^A, \alpha_4^A$, and $\beta_j^A$ represent undetermined coefficients.

Factors affecting the unit yield of grain mainly include variable input elements (fertilizer and labor inputs), fixed capital investment, effective irrigation area, disaster area, and the unit yield of grain in the previous period, etc. Then, the unit yield of grain can be defined as follows:

$$\log Y_{it} = \alpha_0^Y + \alpha_1^Y \log Y_{i,t-1} + \alpha_2^Y \log DIS_{it} + \alpha_3^Y \log N_{it} + \alpha_4^Y \log K_{it}^{al} + \sum_j \beta_j^Y (\log IN_{jt}^s) + \gamma^Y trend + \varepsilon_{it}^Y \quad (2)$$

where $Y_{it}$ represents the unit yield of grain $i$; $DIS_{it}$ represents the disaster area of grain $i$; $N_{it}$ represents the effective irrigation area of grain $i$; $IN_{jt}^s$ represents the variable input element $j$; $K_{it}^{al}$ represents the fixed capital investment of grain $i$. And total grain production ($Q_{it}$) can be expressed as:

$$Q_{it} = A_{it} \times Y_{it} \quad (3)$$

3.1.2. Consumption Module

Grain consumption demand can be divided into food demand, seed production demand, and feed processing demand; among them, the grain consumption of rural residents includes food, seed, and feed, and the grain consumption of urban residents is mainly for food. When constructing grain consumption models, the major factors that affect rural residents' food are, for example, rural per capita disposable income, market grain prices, and related substitute prices. The main factors that affect the food demand of urban residents are, for example, the per capita disposable income of urban residents, the market price of grain, and the price of related substitutes. The main factors affecting the seed production demand and feed processing demand include related substitute prices, grain planting area, and number of livestock raised.

Per capita grain consumption of food demand in rural areas can be expressed as:

$$\log C_{it}^{rural} = \alpha_0^{C1} + \alpha_1^{C1} \log R_t^{rural} + \alpha_2^{C1} \log P_{it}^u + \alpha_3^{C1} \log P_{it}^c + \alpha_4^{C1} \log C_{i,t-1}^{rural} + \alpha_5^{C1} trend + \varepsilon_t^{C1} \quad (4)$$

where $C_{it}^{rural}$ is the per capita grain consumption of food demand in rural areas; $R_t^{rural}$ represents per capita disposable income of rural residents; $P_{it}^c$ represents related substitute price of grain $i$.

Per capita grain consumption of food demand in urban areas can be expressed as:

$$\log C_{it}^{city} = \alpha_0^{C2} + \alpha_1^{C2} \log R_t^{city} + \alpha_2^{C2} \log P_{it} + \alpha_3^{C2} \log P_{it}^c + \alpha_4^{C2} \log C_{i,t-1}^{city} + \alpha_5^{C2} trend + \varepsilon_t^{C2} \quad (5)$$

where $C_{it}^{city}$ is the per capita grain consumption of food demand in urban areas; $R_t^{city}$ represents per capita disposable income of urban residents.

Grain consumption of seed production and feed processing demand can be expressed as:

$$\log C_{it}^{ZS} = \alpha_0^{C3} + \alpha_1^{C3} \log P_{it} + \alpha_2^{C3} \log P_t^{YM} + \alpha_3^{C3} \log Ani_t + \varepsilon_t^{C3} \quad (6)$$

where $C_{it}^{ZS}$ is the grain consumption of seed production and feed processing demand; $P_t^{YM}$ is the market price of maize; $Ani_t$ represents meat production of livestock. Total grain consumption ($D_{it}$) can be expressed as:

$$D_{it} = C_{it}^{rural} \cdot popu_t^{rural} + C_{it}^{city} \cdot popu_t^{city} + C_{it}^{ZS} \quad (7)$$

where $popu_t^{rural}$ represents rural population; $popu_t^{city}$ represents urban population.

### 3.1.3. Stock Module

Grain stock can be divided into the national reserve, social reserve, and farmer's reserve. Farmer's grain storage behavior can be divided into long-term storage behavior and short-term storage behavior. Long-term storage is to protect farmers' food demand, seed production demand, and feed processing demand, while short-term storage is for speculation and profit. According to the grain consumption module, the part of grain consumption demand of rural households has included the self-produced grain consumption demand; thus, the long-term inventory behavior of farmers is no longer included in the expression of the grain stock module. The grain stock in this module mainly refers to the national reserve, social reserve, and farmer's speculative reserve. National reserve is composed of central government special reserve, local government reserve, and general government procurement reserve. The main purpose of the central government special reserve or local government reserve is to ensure the national food security strategy, while the general government procurement reserve is to stabilize the market price of grain, protect the income of grain farmers, and increase the enthusiasm for growing grain. Central government special reserve and local government reserve are of great significance to national food security, and the amount of reserves basically remains stable. China's general government procurement policies include minimum purchase price policy (wheat and rice) and temporary state collection and storage policy (corn). Factors affecting its purchase and storage include changes in grain purchase prices, changes in grain market prices, and changes in grain demand. The goal of social reserve and farmer's speculative reserve is to make profits. Changes in grain purchase prices and changes in grain market prices are the main factors affecting the amount of reserves. At the same time, the general government procurement reserve has a guiding role in social reserve and farmer's speculative reserve. Storage enterprises and farmers will arrange their own storage and sales decisions based on the government's purchasing and storage behavior. Grain stock can be expressed as follows:

$$\log(I_{it}) = \alpha_0^I + \alpha_1^I \log(I_{i,t-1}) + \alpha_2^I \log(P_{it}/P_{i,t-1}) + \alpha_3^I \log(D_{it}/D_{i,t-1}) + \alpha_4^I \log GovI_{it} + \varepsilon_{it}^I \qquad (8)$$

where $I_{it}$ represents the stock of grain $i$, is the sum of national reserve, social reserve, and farmer's reserve; $GovI_{it}$ represents the quantity of national policy storage(general government procurement reserve of wheat).

### 3.1.4. Trade Module

In the production module, $Q_{it}$ comprises domestic consumption and storage and grain exports. In the consumption module, $D_{it}$ comprises domestic production and storage and grain imports. National trade policy is the most direct control factor for grain imports and exports. Tariffs and quotas are the most important grain trade policies. The grain import tariff quota policy requires a low tax rate for imported grain within the quota, and a high tax rate for imported grain outside the quota. From 2002 to 2017, the import tariff rate and import tariff quota for wheat were relatively stable, especially after 2004. The import tariff rate for wheat within the most-favored nation quota was 1%, out of quota was 65%, the ordinary tariff rate was 180%, and wheat tariff quota was 9.636 million tons. After the global food crisis broke out in 2008, China implemented a wheat export tax policy from 2008 to 2009, in order to ensure domestic food supply and reduce food exports. This resulted in a sharp drop in wheat exports in 2008, and imports exceeded exports. In addition, the ratio of domestic and foreign grain prices, exchange rate changes, and grain production are also important factors affecting grain imports and exports.

Grain import can be expressed as:

$$\begin{aligned} \log IM_{it} &= \alpha_0^{IM} + \alpha_1^{IM} \log(P_{it}/P_{it}^f) + \alpha_2^{IM} \log Q_{i,t-1} + \alpha_3^{IM} \log(D_{it}/D_{i,t-1}) + \alpha_4^{IM} \log IM_{i,t-1} + \alpha_5^{IM} \log quo_{it} \\ &\quad + \alpha_6^{IM} \log exra_t + \varepsilon_{it}^{IM} \end{aligned} \qquad (9)$$

Grain export can be expressed as:

$$\log EX_{it} = \alpha_0^{EX} + \alpha_1^{EX} \log(P_{it}/P_{it}^f) + \alpha_2^{EX} \log EX_{i,t-1} + \alpha_3^{EX} tar_{it} + \alpha_4^{EX} \log exra + \varepsilon_{it}^{EX} \tag{10}$$

where $IM_{it}$ is the import of grain $i$; $EX_{it}$ is the export of grain $i$; $P_{it}^f$ represents the international market price of grain $i$; $quo_{it}$ represents the tariff-rate quota for grain import; $exra_t$ is exchange rate; $tar_{it}$ represents export tariff of grain $i$.

### 3.1.5. Price Transfer Module

With China's economic growth and social development, the financial attribute of grain has gradually emerged, and the impact of grain futures price on the market price has gradually increased. Additionally, the grain purchase price, related product price, transportation cost, policy auction price, and grain international price have a certain transmission effect on grain market prices. Thus, the price transfer equation of grain market can be expressed as follows:

$$\log P_{it} = \alpha_0^P + \alpha_1^P \log P_{it}^c + \alpha_2^P \log P_{it}^f + \alpha_3^P \log P_{it}^{fu} + \alpha_4^P \log P_{it}^{bid} + \alpha_5^P \log P_{it}^u + \alpha_6^P \log P_{it}^{tr} + \varepsilon_{it}^P \tag{11}$$

where $P_{it}^{fu}$ is the futures price of grain $i$; $P_{it}^{tr}$ represents the transportation cost; $P_{it}^{bid}$ is the policy auction price of grain $i$.

### 3.1.6. Market-Clearing Condition

The total demand comprises domestic grain consumption, grain stock change, and grain exports. The total supply comprises domestic grain production and grain imports. Thus, the market-clearing condition of grain can be expressed as follows:

$$C_{it}^{rural} \cdot popu_t^{rural} + C_{it}^{city} \cdot popu_t^{city} + C_{it}^{ZS} + (I_{it} - I_{i,t-1}) + EX_{it} = A_{it} \cdot Y_{it} + IM_{it} \tag{12}$$

### 3.1.7. Analysis Framework of the Market-Clearing Condition of the Partial Equilibrium Model

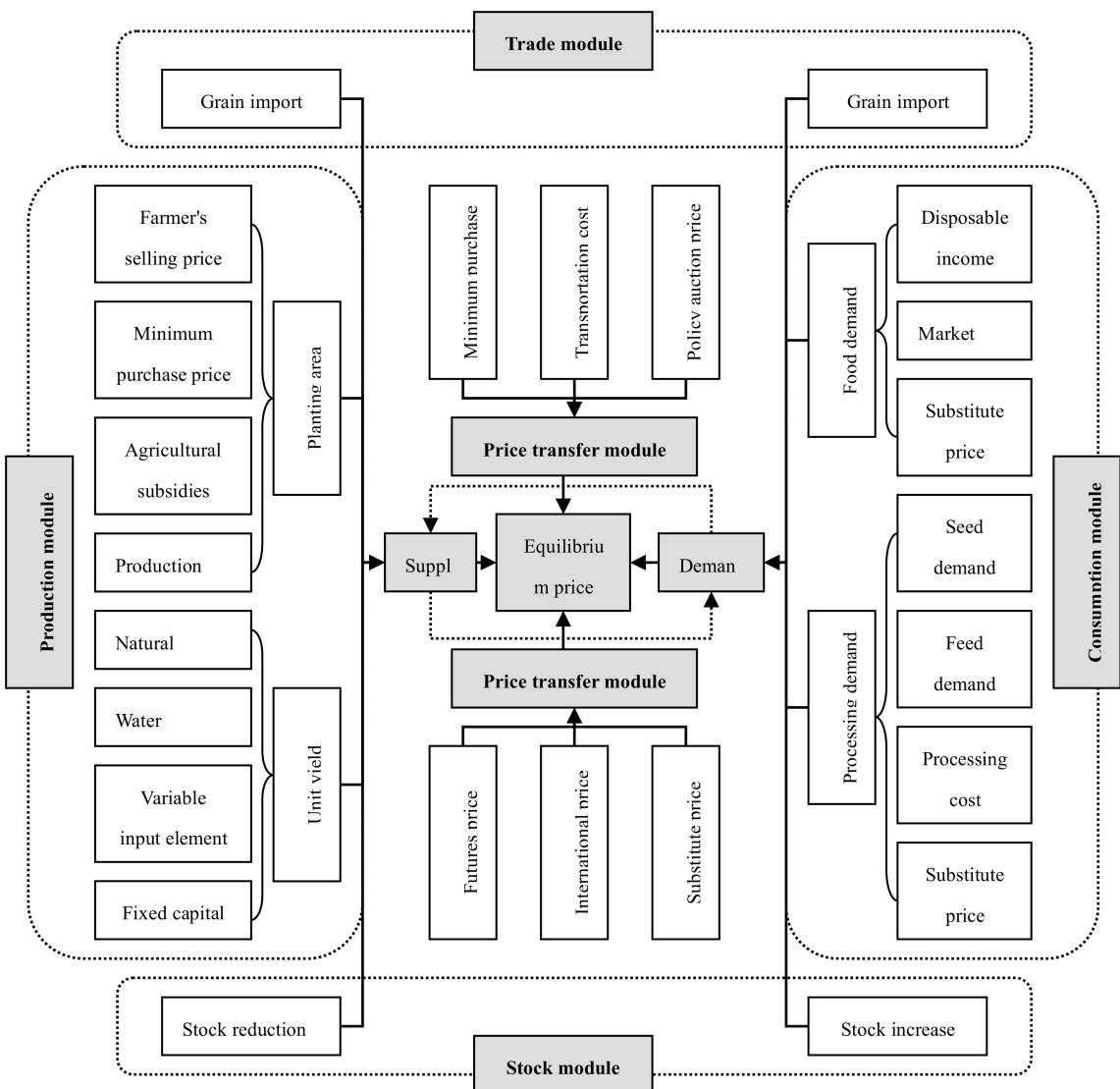

**Figure 1.** Analysis framework of the partial equilibrium model.

### 3.2. Data

Wheat is one of the world's most important grains and one of the three major staple foods in China. Since China implemented its minimum purchase price policy for wheat, the domestic market price has risen rigorously, and stocks and imports have continuously increased. The minimum purchase price policy has caused a serious negative impact on the domestic wheat market; thus, this policy in urgent need of reform. Therefore, studying the reform method of the minimum purchase price of wheat is a representative case for exploring the reform of grain price support policy. In this paper, wheat is the research object when constructing the partial equilibrium model of the grain price mechanism. Because wheat and rice have a mutual substitution effect in terms of food consumption, this paper uses indica rice and japonica rice as wheat substitutes and their market prices as the price of the substitutes. This paper selects domestic diesel retail price ($P_t^{tr}$) data to examine the impact of transportation cost changes. The unit planting cost of wheat ($P_{WH,t}^s$) is used to comprehensively examine the impact of price changes of various production input factors on grain prices. The international price of wheat ($P_{WH,t}^f$) is replaced by the American winter wheat (crusty) FOB price. The fixed capital investment

$(K_t^{al})$ in grain production is replaced by the part of rural fixed assets used for agricultural production. As the state-owned enterprises are the main purchasing and storage entities in national policy-oriented grain purchasing and storage, the state-owned enterprises' grain purchasing and storage can be used to approximately measure the changes in national policy storage $(GovI_{WH,t})$ (Table 1). According to the availability of variable data in this paper, the sample interval is 2000–2017. In order to eliminate the impact of inflation, this paper uses the annual CPI fixed base index based on 2000 to deflate the various types of grain market price, purchase price, minimum purchase price, futures price, policy auction price, production cost, transportation cost, Agricultural production subsidies, residents' income, and capital investment.

**Table 1.** Variables and data sources.

| Variable Name | Symbol | Data Source |
|---|---|---|
| Wheat market price | $P_{WH,it}$ | Data from the Ministry of Agriculture and Rural Affairs of China |
| Farmer's selling price of wheat | $P_{WH,t}^u$ | |
| Indica rice market price | $P_{XH,t}$ | |
| Japonica rice market price | $P_{JH,t}$ | |
| Maize market price | $P_t^{YM}$ | |
| Minimum purchase price | $P_{WH,t}^{gov}$ | Data from news, which released by the Ministry of finance of China |
| International price of wheat | $P_{WH,t}^f$ | Data from the Wind database |
| Wheat futures prices | $P_{WH,t}^{fu}$ | |
| Policy auction price | $p_{WH,t}^{bid}$ | |
| Domestic diesel retail price | $P_t^{tr}$ | |
| Year-end wheat stock | $I_{WH,t}$ | |
| Total wheat consumption | $D_{WH,t}$ | |
| Grain consumption of seed production and feed processing | $C_{WH,t}^{ZS}$ | |
| Unit planting cost of wheat | $P_{WH,t}^s$ | Data from the Compilation of Cost-Benefit Data of Agricultural Products in China (2000–2017) |
| Chemical fertilizer input for wheat production | $IN_{WH,t}^{s(chf)}$ | |
| Labor input for wheat production | $IN_{WH,t}^{s(lab)}$ | |
| Grain-related production subsidies | $SUP_t$ | Data from China Grain Yearbook (2000–2017) |
| Quantity of national policy storage | $GovI_{WH,t}$ | |
| Wheat planting area | $A_{WH,t}$ | |
| Wheat unit yield | $Y_{WH,t}$ | |
| Total wheat production | $Q_{WH,t}$ | |
| Grain disaster rate | $DIS_t$ | Data from the National Bureau of Statistics of China |
| Effective irrigation area | $N_t$ | |
| Per capita disposable income of rural households | $R_t^{rural}$ | |
| Per capita disposable income of urban households | $R_t^{city}$ | |
| Rural population | $popu_t^{rural}$ | |
| Urban population | $popu_t^{city}$ | |
| Fixed capital investment | $K_t^{al}$ | Data from China Rural Statistical Yearbook (2000–2017). |
| Per capita grain consumption of rural households | $C_{WH,t}^{rural}$ | |
| Per capita grain consumption of Urban households | $C_{WH,t}^{city}$ | |

**Table 1.** *Cont.*

| Variable Name | Symbol | Data Source |
| --- | --- | --- |
| Meat production of livestock | $Ani_t$ | |
| Wheat import | $I_{WH,t}$ | Data from the China Customs Database |
| Wheat export | $EX_{WH,t}$ | |
| Exchange rate | $exra_t$ | Data from the website of the people's Bank of China |
| Wheat import tariff-rate quota | $quo_{WH,t}$ | Data from the Customs Import and Export Tariff of China (2000–2017) |
| Export tariff | $tar_{WH,t}$ | |

### 3.3. Analysis Methods

By establishing a local grain model, the endogenous variables in the model system can be obtained including: Wheat market price ($P_{WH,it}$), farmer's selling price of wheat ($P_{WH,t}^u$), wheat planting area ($A_{WH,t}$), wheat unit yield ($Y_{WH,t}$), wheat stock ($I_{WH,t}$), per capita grain consumption of rural households ($C_{WH,t}^{rural}$), per capita grain consumption of urban households ($C_{WH,t}^{city}$), grain consumption of seed production and feed processing ($C_{WH,t}^{ZS}$), import ($I_{WH,t}$), and export ($EX_{WH,t}$). In each modular equation, these endogenous variables can be used not only as dependent variables, but also as independent variables. Therefore, when estimating the partial equilibrium model of wheat, considering the endogeneity of the equations and the contemporaneous correlation of random error terms, three-stage least squares is used to estimate the parameters of the partial equilibrium equation system of wheat [50]. Through the estimation results of the equation parameters in the wheat partial equilibrium model, to simulate the effects of gradually reducing the minimum purchase price, only canceling the minimum purchase price policy, and a combination of canceling the minimum purchase price policy and greatly increasing agricultural subsidies on wheat market price, total production, total consumption, stock, and net import, can we explore the impact of various policy adjustment programs on the sustainable development of China's grain market.

## 4. Analysis Results

### 4.1. Wheat Partial Equilibrium Model Results Analysis

In the wheat planting area equation, the minimum purchase price has the greatest impact on the current wheat planting area, indicating that the planting decision of farmers is greatly affected by the minimum purchase price. The lagged planting area and the lagged purchase price have a greater impact on the current wheat planting area, indicating that the decision-making behavior of farmers to plant wheat is also greatly affected by the previous planting behavior and the lagged purchase price and also reflects the behavioral inertia of farmers when planting wheat. The positive effect of the agricultural subsidy policy on the wheat planting area is small, which reflects that China's agricultural subsidy policy provides little incentive for farmers' enthusiasm for planting at this stage. In recent years, the cost of wheat planting has gradually increased, but because of the implementation of the minimum purchase price policy and grain direct subsidy policy, the wheat planting area has had a fluctuating upward trend; thus, the wheat planting cost has a small negative impact on the planting area (Table 2).

**Table 2.** Estimation results of equation parameters in the wheat partial equilibrium model.

| | | | | | | | | |
|---|---|---|---|---|---|---|---|---|
| **Wheat planting area equation ($A_{WH,t}$)** | | | | | | | | |
| **Variable** | $P^u_{WH,t-1}$ | $P^{gov}_{WH,t}$ | $SUP_t$ | $P^s_{WH,t}$ | $A_{WH,t-1}$ | Intercept | | $R^2$ |
| **Parameter** | 0.103 ** | 0.157 *** | 0.025 * | −0.059 ** | 0.108 * | 4.775 *** | | 0.92 |
| **T-Statistic** | 2.148 | 3.576 | 1.814 | −2.128 | 1.762 | 6.847 | | |

| | | | | | | | | |
|---|---|---|---|---|---|---|---|---|
| **Wheat yield equation ($Y_{WH,t}$)** | | | | | | | | |
| **Variable** | $Y_{WH,t-1}$ | $DIS_t$ | $N_t$ | $IN^{s(chf)}_{WH,t}$ | $IN^{s(lab)}_{WH,t}$ | $K^{al}_t$ | trend | Intercept | $R^2$ |
| **Parameter** | 0.342 * | −0.028 | 0.109 * | 0.155 * | −0.336 *** | 0.065 *** | 0.082 *** | 0.864 *** | 0.95 |
| **T-Statistic** | 1.697 | −1.165 | 1.749 | 1.771 | −7.180 | 6.505 | 5.086 | 4.078 | |

| | | | | | | | |
|---|---|---|---|---|---|---|---|
| **Rural per capita wheat consumption equation ($C^{rural}_{WH,t}$)** | | | | | | | |
| **Variable** | $R^{rural}_t$ | $P^u_{WH,t}$ | $P_{XH,t}$ | $P_{JH,t}$ | $C^{rural}_{WH,t-1}$ | trend | Intercept | $R^2$ |
| **Parameter** | −0.117 ** | −0.139 ** | 0.019 | 0.053 * | 0.149 * | −0.013 | 2.068 *** | 0.87 |
| **T-Statistic** | −2.057 | −2.103 | 0.946 | 1.677 | 1.726 | −1.581 | 4.255 | |

| | | | | | | | |
|---|---|---|---|---|---|---|---|
| **Urban per capita wheat consumption equation ($C^{city}_{WH,t}$)** | | | | | | | |
| **Variable** | $R^{city}_t$ | $P_{WH,t}$ | $P_{XH,t}$ | $P_{JH,t}$ | $C^{city}_{WH,t-1}$ | trend | Intercept | $R^2$ |
| **Parameter** | 0.163 *** | −0.078 ** | 0.125 *** | 0.002 * | 0.448 *** | −0.052 *** | 0.578 *** | 0.90 |
| **T-Statistic** | 4.858 | −2.010 | 3.418 | 1.835 | 5.208 | −1.338 | 3.162 | |

| | | | | |
|---|---|---|---|---|
| **Wheat seed and feed processing consumption equation ($C^{ZS}_{WH,t}$)** | | | | |
| **Variable** | $P_{WH,t}$ | $P^{YM}_t$ | $Ani_t$ | Intercept | $R^2$ |
| **Parameter** | −1.470 * | 0.162 *** | 0.483 * | 2.079 | 0.73 |
| **T-Statistic** | −1.806 | 2.989 | 1.685 | 0.832 | |

| | | | | | |
|---|---|---|---|---|---|
| **Wheat stock equation ($I_{WH,t}$)** | | | | | |
| **Variable** | $I_{WH,t-1}$ | $P_{WH,t}/P_{WH,t-1}$ | $D_{WH,t}/D_{WH,t-1}$ | $GovI_{WH,t}$ | Intercept | $R^2$ |
| **Parameter** | 0.557 *** | −0.438 * | −0.284 | 0.373 ** | 1.107 *** | 0.79 |
| **T-Statistic** | 5.739 | −1.784 | −1.155 | 2.433 | 4.542 | |

| | | | | | | |
|---|---|---|---|---|---|---|
| **Wheat import equation ($IM_{WH,t}$)** | | | | | | |
| **Variable** | $P_{WH,t}/P^f_{WH,t}$ | $Q_{WH,t-1}$ | $D_{WH,t}/D_{WH,t-1}$ | $IM_{WH,t-1}$ | $quo_{WH,t}$ | $exra_t$ | $R^2$ |
| **Parameter** | 5.898 *** | −1.370 *** | 0.587 | 0.245 ** | 16.251 * | −11.650 *** | 0.87 |
| **T-Statistic** | 8.865 | −5.933 | 0.754 | 2.359 | 1.773 | −4.197 | |

| | | | | | |
|---|---|---|---|---|---|
| **Wheat export equation ($EX_{WH,t}$)** | | | | | |
| **Variable** | $P_{WH,t}/P^f_{WH,t}$ | $EX_{WH,t-1}$ | $tar_{WH,t}$ | $exra_t$ | Intercept | $R^2$ |
| **Parameter** | −1.320 *** | −0.216 * | −2.380 * | 6.096 *** | −8.201 *** | 0.84 |
| **T-Statistic** | −4.137 | −1.750 | −1.922 | 4.395 | −3.044 | |

| | | | | | | | | |
|---|---|---|---|---|---|---|---|---|
| **Wheat price transfer equation ($P_{WH,t}$)** | | | | | | | | |
| **Variable** | $P_{XH,t}$ | $P_{JH,t}$ | $P^f_{WH,t}$ | $P^{fu}_{WH,t}$ | $P^{bid}_{WH,t}$ | $P^u_{WH,t}$ | $P^{tr}_t$ | Intercept | $R^2$ |
| **Parameter** | 0.095 * | 0.184 *** | 0.075 *** | −0.191 *** | −0.049 ** | 0.796 *** | 0.083 *** | −0.098 *** | 0.94 |
| **T-Statistic** | 1.912 | 4.885 | 2.569 | −4.433 | −2.054 | 25.139 | 2.796 | −4.855 | |

Notes: * Significant at the 0.1 level, ** Significant at the 0.05 level, *** Significant at the 0.01 level, $i = WH$ represents wheat variety of grain.

In the wheat yield equation, the lagged wheat yield has the greatest positive effect on the current wheat yield, and the amount of chemical fertilizer input has the second-largest impact on the wheat yield. The effect of fixed capital input on wheat yield is less than the effect of chemical fertilizer input. Labor input has a large negative impact on wheat yield, and labor input has a negative correlation with the mechanization level of grain planting. The higher the labor input, the lower the agricultural mechanization level. The level of agricultural mechanization has a great positive impact on grain yield; thus, the labor input has a great negative impact on grain yield. The agricultural disaster area has a small negative impact on wheat yield, and this negative effect is not significant, reflecting that the improvement of agricultural science and technology level since the new century has reduced the impact of natural disasters on the yield of wheat (Table 2).

In the rural per capita wheat consumption equation, the lagged rural per capita wheat consumption has the greatest positive impact on the current rural per capita wheat consumption, indicating that rural per capita wheat consumption is mainly affected by previous consumption habits. The purchase price of wheat has a negative effect because the increase in the wheat purchase price promotes the enthusiasm of farmers to sell wheat. Under the premise of satisfying household consumption, farmers sell as much grain as possible, which leads to a decline in the per capita wheat consumption of rural households. As the income level of the peasant household increases, the expectation of self-retained

grain is adjusted. More farmers gradually adapt to the lifestyle of selling unprocessed grain and buying processed grain. Therefore, the increase in rural per capita disposable income has led to a decline in the per capita wheat consumption of rural households (Table 2).

In the urban per capita wheat consumption equation, lagged urban per capita wheat consumption has the greatest positive impact on current urban per capita wheat consumption, which also shows that rural per capita wheat consumption is mainly affected by previous consumption habits. The wheat market price only has a small negative impact on the urban per capita wheat consumption equation, reflecting the rigidity of the wheat consumption of urban residents. The increase in wheat prices cannot cause a significant reduction in urban wheat consumption (Table 2).

In the consumption equation of wheat seed production and feed processing, the meat production of livestock has a large positive impact, the wheat market price has a large negative impact, and the maize market price has a small positive impact. The difference between the impact of the wheat market price and maize market price is because of the difference in quantities used for feed processing. The main use of maize is for feed, and the main use of wheat is for food. Therefore, when the wheat market price rises, the wheat consumption of feed processing decreases significantly. When the maize market price rises, the consumption of barley, sorghum, and other feed substitute grains increases significantly, but the pulling effect on wheat feed consumption is smaller (Table 2).

In the wheat stock equation, the lagged wheat stock has the largest positive impact on the current wheat stock, indicating that the storage behavior in the previous period has a greater impact on the storage decision in this period. The national policy of purchasing and storing has a substantial positive impact on current wheat stocks. When the wheat minimum purchase price plan was implemented, the quantity of national policy storage increased significantly, leading to a substantial increase in the total wheat stock. Wheat price fluctuation can significantly change wheat stock. The wheat stock includes a large amount of private enterprise reserve, which pursues the principle of profit maximization. When the market price of wheat rises, private enterprises sell wheat in large quantities, leading to a decrease in total wheat stock. The reason that the coefficient of domestic demand change is not significant may be that domestic wheat demand is relatively stable and the change is small in recent years. Compared with national policy storage, domestic demand change is not the main factor causing changes in the wheat stock (Table 2).

In the wheat import equation, the wheat import tariff-rate quota has the largest positive impact on wheat imports, followed by the domestic and international price ratio of wheat. The exchange rate can have the largest negative impact on wheat imports, followed by lagged wheat production. Thus, wheat imports are mainly affected by national food trade policies, exchange rate levels, and the domestic and international price ratio of wheat. This finding shows that wheat imports are mainly affected by national grain trade policy, exchange rate level, and the domestic and international price ratio of wheat (Table 2).

In the wheat export equation, the exchange rate level has the largest positive impact on wheat exports, and the export tariff has a large negative impact on wheat exports. From 2005 to 2007, when the Chinese government implemented the wheat export tax rebate policy, the wheat export volume increased significantly. To restrict the wheat export and ensure domestic supply, China imposed a 3% export tariff in 2008 and 2009, which caused a significant decline in wheat exports (Table 2).

In the wheat price transfer equation, the indica rice market price, japonica rice market price, wheat international price, wheat purchase price, and transportation cost have a positive impact on the wheat market price. The wheat futures price and policy auction price have a small negative impact on the wheat market price. The reason for the negative impact of the wheat futures price might be that China's agricultural futures market is immature, and the wheat futures price cannot effectively reflect the changing trend of the wheat spot price. The reason for the negative impact of the wheat policy auction price is that it is always lower than the market price. With the increase of wheat auction volume, a large quantity of national policy storage of wheat with a low price flows into the market, reducing the overall price level of the wheat market; thus, the policy auction price has a negative impact (Table 2).

## 4.2. Wheat Price Policy Adjustment Simulation and Result Analysis

### 4.2.1. Fit Test of Wheat Partial Equilibrium Model

Through the estimation results of the equation parameters in the wheat partial equilibrium model, to simulate the effects of gradually reducing the minimum purchase price, only canceling the minimum purchase price policy, and a combination of canceling the minimum purchase price policy and greatly increasing agricultural subsidies on wheat market price, total production, total consumption, stock, and net import. Because the out-sample data of the variables are not available, only the in-sample forecast can be performed on the model. The in-sample forecast scheme designed in this paper is as follows: Utilize the estimation results of equation parameters (2000–2017) to make in-sample forecasts for policy adjustments from 2013 to 2017. Before the simulation, the fit test of the wheat partial equilibrium model must be performed [50]. The selected indicators mainly include: Mean absolute error (MAE), mean relative error (MRE), root mean squared error (RMSE), and Theil inequality coefficient (THU). The fit test results of each equation in the wheat partial equilibrium model are shown in Table 2. The smaller the value of each test indicator, the higher the fit degree of the equation. It can be concluded that except for the larger MPE value of the wheat import equation, the value of each test indicator of the other equations is smaller, indicating that the wheat partial equilibrium model has a higher degree of fit, and the next simulation analysis can be performed (Table 3).

**Table 3.** Fit test results of wheat partial equilibrium model.

| Equations | MAE Value | MPE Value | RMSE Value | THU Value |
|---|---|---|---|---|
| Wheat planting area equation | 0.0380 | 0.0069 | 0.0480 | 0.0044 |
| Wheat yield equation | 0.0244 | 0.0162 | 0.0320 | 0.0104 |
| Rural per capita wheat consumption equation | 0.0389 | 0.0179 | 0.0467 | 0.0107 |
| Urban per capita wheat consumption equation | 0.0258 | 0.0160 | 0.0290 | 0.0087 |
| Wheat seed and feed processing consumption equation | 0.1186 | 0.0246 | 0.1590 | 0.0167 |
| Wheat stock equation | 0.1046 | 0.0163 | 0.1428 | 0.0114 |
| Wheat import equation | 1.3656 | 8.7486 | 1.7181 | 0.2531 |
| Wheat export equation | 0.3461 | 0.4026 | 0.4177 | 0.1188 |
| Wheat price transfer equation | 0.0272 | 0.1261 | 0.0341 | 0.0424 |

Mean absolute error (*MAE*), mean relative error (*MRE*), root mean squared error (*RMSE*), and Theil inequality coefficient (*THU*) can be expressed as follows:

$$MAE = \frac{1}{h} \sum_{t=T+1}^{T+h} |\hat{y}_t - y_t| \tag{13}$$

$$MRE = \frac{1}{h} \sum_{t=T+1}^{T+h} \left| \frac{\hat{y}_t - y_t}{y_t} \right| \tag{14}$$

$$RMSE = \sqrt{\frac{1}{h} \sum_{t=T+1}^{T+h} (\hat{y}_t - y_t)^2} \tag{15}$$

$$THU = \sqrt{\frac{1}{h} \sum_{t=T+1}^{T+h} (\hat{y}_t - y_t)^2} \Big/ \left( \sqrt{\frac{1}{h} \sum_{t=T+1}^{T+h} \hat{y}_t^2} + \sqrt{\frac{1}{h} \sum_{t=T+1}^{T+h} y_t^2} \right) \tag{16}$$

where $\hat{y}_t$ represents fitted value of wheat partial equilibrium model; $y_t$ represents actual value of the variable; $h$ is the forecast interval.

4.2.2. Scenario Setting of Wheat Price Policy Adjustment Simulation

Considering that the minimum purchase price policy of grain has been implemented in China for a long time, the grain market price is greatly affected by the policy price. During the marketization reform of the grain price formation mechanism, the impact of the minimum purchase price policy should be gradually reduced. Corresponding subsidy policies should be supported to prevent large fluctuations in grain prices and effectively guarantee farmers' income and enthusiasm for grain production [51]. Therefore, when designing the simulation scheme, policymakers should refer to the design ideas of Zhu et al. [52] and Cao et al. [53]: Assume that the minimum grain purchase price decreases slightly each year; assume that the minimum purchase price has dropped significantly, and it can only compensate for the cost of planting grain; and assume that the minimum grain purchase price has been canceled and that the grain market price was completely determined by market supply and demand. Four simulation schemes are designed in this paper:

Scenario 1. According to the mean value of the rice minimum purchase decline from 2015 to 2017 (as China gradually reduced the rice minimum purchase price from 2015, the mean value of the rice minimum purchase price declined between 2015 and 2017 by 1.39%), the wheat minimum purchase price was set to decrease by 1.39%, 2.78%, 4.17%, 5.56%, and 6.95% from 2013 to 2017 (based on the minimum purchase price in 2012).

Scenario 2. According to the national minimum purchase price standard (the minimum purchase price was announced before the wheat was planted, and the government released the execution plan for the minimum purchase price based on the wheat production cost and the reasonable profit for planting wheat (15–20% of the price). This paper dealt with the reasonable profit as 20% of the price.), the minimum purchase price is set during the simulation period to be reduced to only compensate for the cost of growing grain (the minimum purchase price was decreased by 20%). That is, the minimum purchase price of wheat was reduced by 4%, 8%, 12%, 16%, and 20% from 2013 to 2017 (based on the minimum purchase price in 2012).

Scenario 3. Since 2013, the wheat minimum purchase price policy has been canceled, and the market price of wheat has been completely determined by market supply and demand.

Scenario 4. Since 2013, the wheat minimum purchase price policy has been canceled, and the number of agricultural subsidies has been greatly increased. From 2013 to 2017, the number of agricultural subsidies has increased by 20%, 40%, 60%, 80%, and 100% (based on the amount of agricultural subsidies in 2012).

4.2.3. Simulation Results of Wheat Price Policy Adjustment

By substituting the scenarios into the model, the fitted values of the wheat market price, total production, total consumption, stock, and net import is obtained. By comparing this with the actual values, the impact of the policy adjustment on the wheat market can be obtained. The simulation results of the wheat price policy adjustment are presented in Table 4.

**Table 4.** Change rate of fitted value in simulation of wheat partial equilibrium model.

| Simulation Scheme | Item | 2013 (%) | 2014 (%) | 2015 (%) | 2016 (%) | 2017 (%) |
|---|---|---|---|---|---|---|
| Scenario 1 | Wheat market price | −0.07 | −0.09 | −0.11 | −0.13 | −0.21 |
| | Total production of wheat | −0.13 | −0.23 | −0.34 | −0.43 | −0.52 |
| | Total consumption of wheat | 0.13 | −2.02 | 1.50 | 1.23 | 0.72 |
| | Wheat stock | 0.88 | −3.64 | −3.12 | 2.02 | −3.14 |
| | Net import of wheat | −7.82 | −9.32 | −10.08 | −9.47 | −11.49 |
| Scenario 2 | Wheat market price | −0.13 | −0.63 | −1.60 | −2.18 | −2.63 |
| | Total production of wheat | −1.89 | −2.85 | −3.36 | −3.84 | −4.38 |
| | Total consumption of wheat | 2.47 | −0.72 | 2.88 | −0.22 | 1.34 |
| | Wheat stock | 7.52 | −5.04 | −4.32 | 6.44 | −7.08 |
| | Net import of wheat | −9.58 | −11.10 | −17.13 | −16.15 | −17.90 |
| Scenario 3 | Wheat market price | −9.34 | −12.28 | −14.23 | −11.60 | −9.21 |
| | Total production of wheat | −7.61 | −8.63 | −8.67 | −8.56 | −8.10 |
| | Total consumption of wheat | 2.83 | 0.86 | 3.44 | 1.30 | 3.03 |
| | Wheat stock | −55.15 | −55.73 | −53.39 | −50.45 | −46.37 |
| | Net import of wheat | −51.51 | −56.16 | −70.30 | −64.65 | −59.37 |
| Scenario 4 | Wheat market price | −9.12 | −10.96 | −10.52 | −8.58 | −6.26 |
| | Total production of wheat | −6.50 | −7.29 | −6.85 | −6.37 | −5.83 |
| | Total consumption of wheat | 1.35 | −1.03 | 1.64 | 1.43 | 2.04 |
| | Wheat stock | −50.89 | −51.24 | −47.65 | −42.54 | −38.98 |
| | Net import of wheat | −50.91 | −54.92 | −57.61 | −46.73 | −40.43 |

Notes: the rate of change = $(\hat{y}_t - y_t)/y_t$; total wheat production = $A_{WH,t} \cdot Y_{WH,t}$; wheat net import = $(IM_{WH,t} - EX_{WH,t})$; total wheat consumption = $\left(C_{WH,t}^{rural} \cdot popu_t^{rural} + C_{WH,t}^{city} \cdot popu_t^{city} + C_{WH,t}^{ZS}\right)$.

From the simulation results for Scenario 1, after the wheat minimum purchase price was decreased by 1.39%, 2.78%, 4.17%, 5.56%, and 6.95% from 2013 to 2017, respectively, the wheat market price and total production showed a slight downward trend by year, and the total consumption and stock showed a small fluctuation trend; the net import decrease was relatively large. A small decrease in the wheat minimum purchase price caused a small negative impact on market price and total production, a large negative impact on net import, and a small positive and negative impact on total consumption and stock (Table 4).

From the simulation results of Scenario 2, after the wheat minimum purchase price was decreased by 4%, 8%, 12%, 16%, and 20% from 2013 to 2017, respectively, it has caused a small negative impact on market prices and total production, a small positive impact on total consumption, a positive and negative impact on stock, and a large negative impact on net import (Table 3).

From the simulation results of Scenario 3, after the cancellation of the minimum purchase price policy, wheat stock, and net import have fallen by a huge margin, the wheat market price and total production have fallen by a relatively large margin, and total consumption has increased by a small margin (Table 4).

From the simulation results of Scenario 4, after the cancellation of the minimum purchase price policy, and the amount of agricultural subsidies increased by 20%, 40%, 60%, 80%, and 100% from 2013 to 2017, respectively, the change rate of the wheat market price, total production, total consumption, stock, and net import has declined compared with Scenario 3, and the overall level of change remains relatively high (Table 4).

## 5. Discussion

Combined with the analysis of the estimation results of the wheat partial equilibrium model, the simulation results under different scenarios can be explained. The minimum purchase price of wheat affects the supply and demand of the wheat market by affecting total production, and then affects the farmer's selling price and market price. After the farmer's purchase price and market price of wheat change, there are feedback regulations regarding total production, total consumption, stock, and net import. In the wheat partial equilibrium model, the estimated values of the wheat market price (or farmer's selling price) coefficients in the production, consumption, and stock equations are smaller, and the estimated value of the wheat market price coefficient in the import equation is large.

Additionally, China's grain import quota policy has resulted in a small proportion of net wheat imports in China's total wheat production. Therefore, when the domestic market price falls, the net wheat import decreases significantly.

In Scenario 1, when the minimum purchase price has decreased by a small amount, the domestic and international price spread of wheat decreased by a small amount, the wheat stock pressure also slightly decreased, but the wheat import pressure decreased by a large amount. However, because of the actual conditions such as large domestic and international price spreads and a large base of policy stocks, a small reduction in the wheat minimum purchase price cannot effectively suppress its negative impacts. Therefore, this policy adjustment has little effect on the healthy development of China's wheat market, and the implementation of this policy adjustment is less sustainable.

In Scenario 2, compared with Scenario 1, the domestic and international wheat price spread, stock, and net import have declined, and this policy adjustment has a less negative impact on total wheat production. Therefore, this policy adjustment has a better impact on the development of China's grain market than Scenario 1. Although the wheat minimum purchase price has been reduced to a level that can only compensate for the cost of planting wheat (20% reduction), the policy has not been canceled; thus, the negative impact of the policy cannot be eliminated from the source.

In Scenario 3, after the cancellation of the minimum purchase price policy, the wheat market price lost the protection of the minimum purchase price policy and was only regulated by the market supply and demand, which caused the simulated price to drop significantly compared with the actual price in the short term. Due to the effects of market rebalancing and feedback regulation, the decline in wheat market prices has further impacts on total production and total consumption. When the minimum purchase price policy was canceled, the national reserve of wheat reduced greatly, and it also had a negative effect on the social reserve. Therefore, in the early stage of canceling the minimum purchase price policy, the simulated stock reduced significantly compared with the actual stock. Due to the market adjustment mechanism, the decrease in the range of market price and total production has gradually declined, and the changes in total consumption and net import tend to be stable; thus, the decline range of wheat stock has decreased. The net import of wheat was greatly affected by the decline in market price, and the cancellation of the minimum purchase price caused a significant decline in the wheat market price, which led to a continuous and substantial decline in net import. When the minimum wheat purchase price policy was canceled, the domestic and international price spreads of wheat and the stock and import of wheat decreased significantly. Due to the loss of the protection of the minimum purchase price policy, wheat production has dropped significantly. Although the negative impact of the minimum purchase price policy has been essentially resolved, it has had a substantial impact on China's grain supply security. Thus, the implementation of this policy adjustment is less sustainable.

In Scenario 4, with the cancellation of the wheat minimum purchase price policy, although the number of agricultural subsidies increased substantially each year, the subsidies had little effect on restraining price fluctuations in the wheat market. Considering the current agricultural subsidies, the direct subsidies for planting wheat, which are closely related to farmers' income, account for a small proportion of the total agricultural subsidies. Although the overall increase in agricultural subsidies is relatively large, the number of direct subsidies allocated to each farmer remains small and cannot significantly increase farmers' enthusiasm for planting grain; thus, the total wheat production has still declined. In this policy adjustment scheme, the domestic and international price spread of wheat has significantly decreased, and the stock and import have greatly decreased. Due to the limited effect of the current agricultural subsidy structure on improving farmers' enthusiasm for growing grain, the total wheat production has still declined, but the decline has significantly improved compared with that in Scenario 3. In summary, Scenario 4 compensates for the limitations of Scenario 3 and not only solves the negative impact of the minimum purchase price policy, but also reduces the impact of the cancellation of the minimum purchase price policy on grain supply security. This policy adjustment is more sustainable than the other scenarios are. However, what remains necessary is to optimize the

structure of agricultural subsidies and increase the efficiency of subsidies to ensure the security of China's grain supply.

## 6. Conclusions and Policy Implications

### 6.1. Conclusions

This paper builds a partial equilibrium model of China's grain market. Based on the analysis of the operation mechanism of China's grain market, the relevant data of China's wheat is used to solve the estimated values of the parameters in the simultaneous equations. Through the estimation values of equation parameters, we simulate the effects of gradually reducing the minimum purchase price, only canceling the minimum purchase price policy, and a combination of canceling the minimum purchase price policy and greatly increasing agricultural subsidies on wheat market price, total production, total consumption, stock, and net import. Next, the sustainability of various adjustment programs was evaluated. The main conclusions are as follows:

1.  On the basis of the estimation results of the equation parameters in the wheat partial equilibrium model, the wheat minimum purchase price and agricultural subsidies have a significant positive impact on the wheat planting area. The wheat market price and the farmer's selling price have significant effects on the wheat planting area, wheat consumption, seed production and feed processing consumption, stock, and net import. When the minimum purchase price or agricultural subsidy changes, the supply and demand relationship in the wheat market is impacted, leading to changes in the wheat market price and the farmer's selling price. Then, it has feedback regulations on production, consumption, stock, and net import, and the adjustment effect of the minimum purchase price is greater.

2.  A small reduction in the minimum purchase price of wheat has little impact on the wheat market price, total production, and total consumption but has a substantial impact on stock and net import. The greater the reduction in the minimum purchase price, the higher the rate of change in wheat market prices, total production, total consumption, stock, and net import. Due to actual conditions such as large domestic and international price spreads and large base of policy stocks, a small reduction in the wheat minimum purchase price cannot effectively suppress its negative impacts, and the implementation of this policy adjustment is less sustainable.

3.  After the cancellation of the minimum wheat purchase price policy, the decline in wheat stock and net import was very large, the decline in wheat market price and total production was relatively large, and the increase in total wheat consumption was small. The wheat market price, production, and stock declined significantly in the early stage, but because of the adjustment effect of the market mechanism, its decline range decreased gradually with the increase in simulation time. The cancellation of the minimum purchase price policy can significantly reduce domestic and foreign price spread, stock, and import but will also significantly reduce wheat production in the short term and threaten China's grain security.

4.  The cancellation of the minimum wheat purchase price and the increase of agricultural subsidies can solve the negative impact of the minimum purchase price policy and reduce the impact of the cancellation of the minimum purchase price policy on grain supply security. This policy adjustment is more sustainable than others. However, the current agricultural subsidy in China is inefficient, and its structure is unreasonable. Therefore, to achieve the healthy development of China's grain market and improve the sustainability of China's grain policy, the efficiency of agricultural subsidies should be improved, and the structure of agricultural subsidies should be optimized.

### 6.2. Policy Implication

The paper explores the influence of different reformation plans of China's wheat grain price policy on China's wheat market price, total production, total consumption, stock, and net import.

The conclusion based on the results reveals the basic direction of reform for China's wheat minimum grain purchase prices and provides insights for the Chinese government, Chinese growers, foreign governments, foreign growers, and institutional investors.

1.  To protect China's food security and fulfill its relevant commitments to agricultural cooperation in international trade negotiations, the Chinese government should formulate a sustainable agricultural policy system consistent with the current situation to realize domestic prices that connect with foreign countries and reduce inventory pressure. In the implementation of the grain market-oriented reform, the minimum purchase price should be gradually reduced by a small margin rather than being canceled at one time. The Chinese government should also optimize the structure of agricultural subsidies and accelerate the shift to "green box" subsidy policies, such as grain planting income subsidies or target price insurance, to ensure the sustainable development of grain security.

2.  The current grain price support policy has increased farmers' selling price, stabilized the income of grain planting, and guaranteed the enthusiasm for grain planting; however, it has also caused serious policy dependence, resulting in farmers' decision-making on planting being largely affected by the grain price support policy rather than directly determined by market supply and demand factors. Conducting the market-oriented reform of the grain price support policy and using market means to adjust the allocation of grain production factors will help farmers make reasonable adjustments to planting decisions based on market conditions. Reasonable subsidies to farmers can also ensure the enthusiasm of grain cultivation. Therefore, the market-oriented reform of the grain price support policy should make the new policy more sustainable.

3.  The market-oriented reform of China's grain price support policy will bring the domestic price of wheat in line with the international price. Additionally, because of the grain import tariff quotas, China's grain market is less speculative. Therefore, foreign institutional investors should return to traditional businesses such as transportation services and risk hedging in the long term. Notably, the reduction in the price gap between home and abroad will make foreign wheat gradually lose price competitiveness in China. Planting high-quality wheat or switching to complementary products is key to maintaining a competitive advantage in the Chinese market for foreign farmers.

4.  In adjusting its grain price support policy, the Chinese government has adopted gradual reform and improved the sustainability of policy implementation. China could simultaneously consider the practice of ensuring food security, increasing farmers' income from planting grain, reducing the gap between the domestic and foreign price, and removing redundant stock, which could provide experience for policymakers of emerging or less developed countries.

**Author Contributions:** Conceptualization, W.L. and J.L.; methodology, J.L. and Z.S.; software, J.L.; validation, J.L.; investigation, J.L.; resources, J.L.; data analysis, J.L.; writing—original draft preparation, J.L.; writing-review and editing, W.L. and Z.S.; visualization, J.L.; supervision, W.L. and Z.S.; project administration, Z.S.; funding acquisition, Z.S. All authors have read and agreed to the published version of the manuscript.

**Funding:** This research was funded by Program of the Natural Science Foundation of China (41871120). Priority Research Program of Chinese Academy of Sciences (XDA20010102).

**Conflicts of Interest:** The authors declare no conflict of interest.

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
