# Peer review of "Sustainability of the Adjustment Schemes in China’s Grain Price Support Policy—An Empirical Analysis Based on the Partial Equilibrium Model of Wheat"

_sustainability, doi:10.3390/su12166447_

Round 1
Reviewer 1 Report
The manuscript is well written and well structured, it relates closely to sustainability issues and includes policy implications.
The supporting references are adequate and recent.
Some acronyms should be explained when they first appear in the text. There are some minor corrections of typos required.
The authors should include in the Discussion Section the future lines of research and the limitations of the present study
Author Response
We want to express our deep thanks to your comments. We have added explanations of relevant agricultural policies, and corrected some errors in expression, and added discussions on the direction of future grain policy reforms.
Reviewer 2 Report
Please see accompanying file

Round 2
Reviewer 2 Report
Many thanks for the detailed responses you have made to my previous comments. I think that this extra explanation of Chinese policy will not only be of interest of itself to readers, but it helps greatly in providing the context against which your results and policy implications can be judged.